# Aerosol Characteristics in the Three Poles of the Earth as Characterized by CALIPSO

Yikun Yang[1,2], Chuanfeng Zhao[1,2], Quan Wang[3], Zhiyuan Cong[4], Xingchun Yang[1,2], Hao Fan[1,2]

[1]College of Global Change and Earth System Science, State Key Laboratory of Earth Surface Processes and Resource Ecology, Beijing Normal University, Beijing 100875, China
[2]Joint Center for Global Change Studies, Beijing Normal University, Beijing 100875, China
[3]Department of Atmospheric Physics, Nanjing University, Nanjing 210046, China
[4]Key Laboratory of Tibetan Environment Changes and Land Surface Processes, Institute of Tibetan Plateau Research, Chinese Academy of Sciences (CAS), Beijing, 100101, China

*Correspondence to: Chuanfeng Zhao (czhao@bnu.edu.cn)*

**Abstract.** To better understand the aerosol properties over the Arctic, Antarctic, and Tibetan Plateau (TP), the aerosol optical properties were investigated using 13 years CALIPSO L3 data, and the back trajectories for air masses were also simulated using the Hybrid Single Particle Lagrangian Integrated Trajectory (HYSPLIT) model. The results show that the aerosol optical depth (AOD) has obvious spatial and seasonal variation characteristics, and the aerosol loading over Eurasia, Ross Sea, and South Asia is relatively large. The annual average AODs over the Arctic, Antarctic, and TP are 0.046, 0.024, and 0.098, respectively. Seasonally, the AOD values are larger from late Autumn to early spring in the Arctic, in winter and spring in the Antactic, and in spring and summer over the TP. There are no significant temporal trends of AOD anomalies in the three study regions. Clean marine and dust-related aerosols are the dominant types over ocean and land respectively in both the Arctic and Antarctic, while dust-related aerosol types have greater occurrence frequency (OF) over the TP. The OF of dust-related and elevated smoke is large for a broad range of heights, indicating that they are likely transported aerosols, while other types of aerosols mainly occurred at heights below 2 km in the Antarctic and Arctic. The maximum OF of dust-related aerosols mainly occurs at 6 km altitude over the TP. The analysis of back trajectories of the air masses shows large differences among different regions and seasons. The Arctic region is more vulnerable to mid-latitude pollutants than the Antarctic region, especially in winter and spring, while the air masses in the TP are mainly from the Iranian Plateau, Tarim Basin, and South Asia.

# 1    Introduction

As an important component, atmospheric aerosols play a crucial role in the Earth-atmosphere system (Garrett and Zhao, 2006; Ghan and Easter, 2006; Nabat et al., 2015; Wei et al., 2021; Xue et al., 2020). Aerosols have a variety of effects on Earth's climate, including the significant direct effect (Rap et al., 2013; Xing et al., 2017), indirect effect (Albrecht, 1989; Liu et al., 2019; 2020a; Righi et al., 2011; Twomey, 1997; Zhao and Garrett, 2015), and semi-direct effect (Amiri-Farahani et al., 2017; Johnson, 2005; Koren et al., 2005). Meanwhile, different aerosol types often have different physical, chemical, and optical properties, and the balance between cooling and warming depends to some extent on aerosol characteristics (Boucher et al., 2013). The influence of aerosols on the Earth-atmosphere system depends on aerosol characteristics and underlying surface (Kipling et al., 2016; McFarlane et al., 2007). The vertical distribution of aerosol is especially valuable as a signature of combined impacts, including the processes of aerosol emission, conversion, transport, and removal (Winker et al., 2013). Due to the lack of understanding of aerosol distribution, dynamics, and optical characteristics, the impact of aerosols on the global radiative budget in climate models has great uncertainty (Boucher et al., 2013; Loeb and Su, 2010). Thus, knowledge of aerosol characteristics is essential for determining the radiative forcing effects of aerosols, improving the accuracy of aerosol optical depth (AOD) retrieval using passive satellites, and quantifying the role of aerosols in global climate changes.

The acquisition of aerosol characteristics is mainly from two methods, ground-based monitoring and satellite remote sensing (Giles et al., 2012; Nishizawa et al., 2007; Omar et al., 2005; Russell et al., 2014). Ground-based remote sensing, such as the aerosol robotic network (AERONET), can provide high accuracy aerosol characteristics. The aerosol properties from AERONET are derived from direct sun extinction and sky radiance measurements, including columnar optical depth, single scattering albedo (SSA), Ångström exponent (AE), and so on (Dubovik and King, 2000; Dubovik et al., 2002; 2006). Although the aerosol characteristics can be obtained from ground-based remote sensing with high accuracy, they have some limitations in the study of global aerosol characteristics research. On one hand, it is difficult to acquire the vertical distributions of aerosols. On the other hand, due to the strong spatiotemporal variations of aerosols, the spatiotemporal representation of aerosol characteristics measured by ground stations is limited.

Passive satellite remote sensing also can be used to obtain aerosol properties. In general, passive remote sensing can only obtain two-dimensional aerosol characteristics, but cannot obtain aerosol vertical

structure information. Several AOD retrieval algorithms based on passive remote sensing have been developed over the past decade, such as Dark Target (DT), Dark Water, Deep Blue (DB), and Multi-Angle Implementation of Atmospheric Correction (MAIAC), structure-function algorithm, and so on (Hsu et al., 2013; Hsu et al., 2004; Kaufman et al., 1997; Levy et al., 2013; Lyapustin et al., 2018; Martonchik et al., 1998; Tanre et al., 1988). In terms of aerosol type, Multi-angle Imaging

SpectroRadiometer (MISR) instrument, which has nine view angles along the flight path (Diner et al., 1998), is sensitive to the size and shape of aerosols (Diner et al., 2008). Ozone Monitoring Instrument (OMI) includes ultraviolet bands, which can be used to retrieve aerosol optical parameters, such as absorbing aerosol optical depth, single scattering albedo, and aerosol index (Marey et al., 2011; Torres et al., 2007). Compared with passive satellite remote sensing, active satellite remote sensing, such as the

Cloud-Aerosol Lidar with Orthogonal Polarization (CALIOP), can acquire the vertical profile of the atmosphere and understand the vertical distribution of aerosol properties at a local or global scale (Shimizu et al., 2016). With three elastic backscattering channels, CALIOP is the first polarization lidar in space to provide three-dimensional atmospheric structure measurements (Granados-Muñoz et al., 2019; Peyridieu et al., 2010). It can measure the vertical distribution, microphysical, and optical properties of

aerosols and clouds with a high vertical resolution at 1064 nm and a parallel and cross-polarized return signal at 532 nm (Kittaka et al., 2011; Kumar et al., 2016). CALIOP has high sensitivity and can detect weak aerosol layers with optical depths of 0.01 or less (Winker et al., 2007). The polarization measurements also allow the discrimination of spherical and non-spherical cloud and aerosol particles. Thus, CALIOP is widely used to the study of aerosol and cloud characteristics (Das and Jayaraman, 2011;

Sun et al., 2018; Varnai and Marshak, 2010).

As two main cold sources of the global atmosphere, the Arctic and Antarctic play an irreplaceable key role in global climate change research. Located in the middle of Asia, the Tibetan Plateau (TP) is the largest ice sheet accumulation area except for the Arctic and Antarctic. The Arctic, Antarctic, and TP are representative of pristine regions, and they are very sensitive to global climate change (Lu et al., 2011).

Associated with their different geographical environments, human activities have different effects on them. Previous studies have indicated that the clouds and radiation are particularly sensitive to aerosols

over the pristine regions (Garrett and Zhao, 2006; Seinfeld et al., 2016; Wang et al., 2018). The Arctic, Antarctic, and TP have been undergoing unprecedented changes in global climate changes.

Extensive researches about aerosol properties over the pristine regions have been conducted (Di Carmine et al., 2005; Leaitch et al., 2020; Wu et al., 2018). The Arctic is a region with ample spatiotemporal variability in aerosols (Schmeisser et al., 2018). Due to the influence of pollutants transported (e.g. forest fire smoke, dust, soot, and sulfates) from lower latitudes, the AOD in the Arctic is abnormally high in winter and spring (Stone et al., 2014; Tomasi et al., 2007). While in the summertime, the oxidation of dimethyl sulfide (DMS), emitted by phytoplankton activity in the marine, can act as cloud condensation nuclei and exert significant control on sulfate aerosol (Leaitch et al., 2013). Meanwhile, by employing carbon monoxide as the assumed passive tracer, the relative contributions of transport efficiency and scavenging to seasonal variability of Arctic aerosol have also been evaluated (Garrett et al., 2010). In the past few decades, the aerosol properties in the Antarctic region, including their concentrations, size distribution, and chemical composition, have been investigated mainly based on ground-based observations (Barbaro et al., 2017; Kerminen et al., 2000; Koponen et al., 2003). The aerosol properties of the Antarctic are mainly controlled by the Southern Ocean primary and secondary emissions and some periodical long-range transport (Asmi et al., 2018). Sea-salt coarse particle and sulfate fine particle aerosols are most abundant in the coastal Antarctic regions and over the Antarctic continental regions, respectively (Hall and Wolff, 1998; Wagenbach et al., 1998; Kerminen et al., 2000). Meanwhile, there are also obvious seasonal differences in Antarctic aerosol types. Sea salt and ammonium sulfate particles are dominant in the polar night months, while sulfuric acid droplets are the main particles in the sunlit months (Ito, 1985). The types of aerosols in the TP are complex, and the dominant aerosol type varies with site (Zhao et al., 2020). Dust aerosols in the northern parts of the TP and polluted aerosols over South Asia can reach internal regions of the TP through long-distance transport (Cong et al., 2015; Huang et al., 2007; Lu et al., 2012; Lüthi et al., 2015; Xia et al., 2011; Zhao at al., 2013; Zhu et al., 2019).

Although many studies have been carried out on aerosol optical properties over the Arctic, Antarctic, and TP, they are mainly based on the short-term ground remote sensing or in-situ observations, which has limited spatial representation (Chaubey et al., 2011; Cong et al., 2009; Eleftheriadis et al., 2004; Engvall et al., 2008; Pokharel et al., 2019), and inadequate information about the vertical distribution of aerosols. Meanwhile, different aerosol types can result in large uncertainty in estimating the aerosol radiative effect

(Loeb and Su, 2010). Thus, it is essential to investigate the long-term aerosol characteristics over relatively large domains of the three pole regions, including the vertical profile information. In this study, the aerosol optical properties over the Arctic, Antarctic, and TP were investigated systematically, including the spatial and temporal distribution, vertical structure, and temporal trends of AOD and aerosol types. In addition, the back trajectory of air masses was also performed to determine the influence of ambient aerosols on the study areas.

## 2 Data and Methods

### 2.1 Study regions

As shown in Figure 1, the Arctic, Antarctic, and TP are selected as our study regions. The areas north of 65° N and south of 65° S are the study regions of the Arctic and Antarctic respectively, as shown in Figure 1 (a) and (b). The Arctic is an ocean covered by a thin layer of perennial sea ice and surrounded by land including Asia, Europe, and North America, while the Antarctic is dominated by the continent covered by a very thick ice cap and surrounded by a rim of sea ice and the Southern Ocean. The TP is composed of land and ice sheets, and the surrounding environment is complex. As shown in Figure 1 (c), there are the Taklimakan Desert in the north and the heavily polluted South Asia in the south. Due to the coarse resolution of CALIPSO L3 data over the TP, the spatial and temporal distributions, as well as the temporal variation trends, were captured in a large region with latitudes from 25° to 41° N and longitudes from 65° to 105° E. However, the vertical characteristics of aerosol properties were only investigated in the inner region of the TP, which is marked by black dots as shown in Figure 1 (c). In addition, eleven special locations (marked with green pentagrams) were selected for the study of aerosol sources using back trajectories, and the detailed information of eleven sites can be found in Section 2.3 and Table S1.

### 2.2 CALIOP data

The CALIPSO satellite provides new sight into the role of how clouds and aerosols form, evolve, and affect weather and climate (Winker et al., 2007; 2010). Level 3 tropospheric aerosol profile product based on level 2 aerosol extinction profiles has the highest quality and is the most sophisticated among all CALIOP level 2 data products (Kim et al., 2018). Compared with the previous products, several changes in data quality screening have been made in the latest product to further avoid extinction retrieval errors, and the detailed algorithm has been depicted (Tackett et al., 2018).

Compared with other sky conditions, the level 3 tropospheric cloud-free aerosol profile (NL3TCFAP) product has the highest quality as extinction retrievals are minimally affected by errors in retrieving the attenuation of overlying cloud cover (Tackett et al., 2018). Meanwhile, the NL3TCFAP product can describe in detail the near-global three-dimensional distribution of aerosols. Thus, to investigate the aerosol properties over three polar regions of the Arctic, Antarctic, and TP, the NL3TCFAP product including day and night time was used in this study. Up to now, the NL3TCFAP product contains seven types of aerosols, which are clean marine, dust, polluted continental/smoke, clean continental, polluted dust, elevated smoke, and dusty marine. The properties of different types of aerosols will be discussed in Section 3.2.

The NL3TCFAP product records aerosol properties data on a uniform 2° latitude by 5° longitude grid, and has a vertical resolution of 60 m for heights up to 12.1 km above mean sea level. In this study, the mean AOD of each grid at different temporal scales was calculated, and the seasonal differences between the northern and southern hemispheres were also considered. The spring (autumn), summer (winter), autumn (spring), and winter (summer) are defined as March-May, June-August, September-November, and December-February in the north (south) hemisphere, respectively. Note that the averaged aerosol properties over the TP region in this study are only for the internal pixels of TP, which is marked by black dots in Figure 1 (c). The occurrence frequency (OF) of aerosol types was also calculated by counting the number of samples of seven aerosol types in each horizontal grid cell or altitude layer. For the vertical distribution of aerosol properties such as extinction coefficient of the dominant type of aerosol, the CALIOP data was used with further data quality control by removing the outliers. The outliers are defined as the observed data ($x$) falling outside three times of the standard deviations ($\delta$) above or below the mean ($\bar{x}$), as follows:

$$x < \bar{x} - 3 \times \delta \ \ or \ \ x > \bar{x} + 3 \times \delta \qquad (1)$$

### 2.3 HYSPLIT model

The Hybrid Single-Particle Lagrangian Integrated Trajectory (HYSPLIT) model has been widely used in the simulation of atmospheric pollutant transport, dispersion, and deposition (Ashrafi et al., 2014; Jeong et al., 2012; Vernon et al., 2018; Zhao et al., 2009). To fully understand the sources of aerosols, the back trajectories of air masses at eleven selected sites over three study regions mentioned above were examined using the latest version (V5.0.0) of the HYSPLIT model (Stein et al., 2015). Simultaneously,

the multiple trajectories that are near each other were merged into groups through cluster analysis. In this study, the four Arctic sites are located in Greenland (N1), Northern Europe (N2), Northern Asia (N3), and Northern North American (N4). The four sites in the Antarctic are located on the Antarctic Peninsula (S1), Ross Sea (S2), Dronning Maud Land (S3), and Wilkes Land (S4). The two selected sites in the TP region are located on the northern (TP1), southern (TP2), and eastern (TP3) edges of the TP region. The locations of these sites are shown in Figure 1, and the detailed information of each site is shown in Table S1. Previous air mass back trajectory simulations in the Polar regions found that it is difficult to simulate the seasonal difference of the air mass with short-term back trajectory simulation, while the long-term back trajectory simulation has great uncertainties in the spatial domain (Hirdman et al., 2010; Sharma et al., 2013), thus a 14-day back trajectory simulation was adopted in this study (Rousseau et al., 2006), and the simulation date was set as the 15th and last day of each month which can help save a lot of computation sources while keeping the simulated back trajectories representative.

## 3    Results and Discussion

### 3.1    The spatial and temporal distribution of aerosol properties

#### 3.1.1 The spatial distribution of AOD

Figure 2 depicted the seasonal averaged spatial distribution of AOD over the Arctic, Antarctic, and TP, from which we can find that the AOD averaged between June 2006 and December 2019 has obvious spatial variation. In the Arctic, except for Greenland Island, aerosol loadings are larger over the continent than that over the ocean. On the contrary, aerosols loadings over the Antarctic continent are lower than that over the surrounding ocean. In general, aerosol loadings are found larger in the southern part of the Atlantic Ocean in the Antarctic and decrease with the increase of latitude, while high AODs could exist in some regions at high latitudes of the Antarctic such as the Antarctic Peninsula, especially in spring and winter. The aerosol concentration in the TP region is generally low while the aerosol loading in the regions around the TP (e.g. Tarim Basin in the north, Qaidam Basin in the northeast, Sichuan Basin in the east, and South Asia in the south) is large. In terms of regional differences, the aerosol concentration in the Arctic region is significantly higher than that in the Antarctic region. Meanwhile, the annual average AODs over the Arctic, Antarctic, and the inner region of the TP are 0.046, 0.024, and 0.098 with the standard deviations of 0.003, 0.002, and 0.009, respectively.

### 3.1.2 The multi-year averaged seasonal variation of AOD

The aerosols and monsoon circulation patterns interact with each other (Ma and Guan, 2018), making it particularly valuable to know the seasonal variations of aerosol properties. In this study, we investigated the monthly variations of multi-year (June 2006 - December 2019) averages and standard deviations of AODs for three study regions, which are shown in Figure 3. As shown in Figures 2 and 3, AOD has obvious seasonal variations, especially over the TP and Arctic, while the Antarctic AOD has relatively weak seasonal variations. The TP has a higher aerosol concentration in spring and summer. The high aerosol concentration mainly occurs from late autumn to early spring in the Arctic, while in winter and spring in the Antarctic. The aerosol loading over the TP is easily affected by the surrounding regions where there are many anthropogenic and natural aerosol sources. Specifically, the dust aerosols in the Tarim Basin and Qaidam Basin have a greater contribution to the TP in spring and summer, especially in the northern part of the TP in summer (Huang et al., 2007; Xia et al., 2008; Xu et al., 2020). Meanwhile, a large number of fine aerosol particles exist in South Asia and the northern Indian Peninsula due to forest fires and anthropogenic burning during the dry season. The aerosols are lifted and transported to the Himalayas under the influence of large-scale atmospheric systems such as the South Asian monsoon and the Siberian high, which affects the southern part of the TP (Cong et al., 2015; Engling et al., 2011; Han et al., 2020; Xu et al., 2014; 2015).

The high aerosol concentration in the winter and spring Arctic is known as the Arctic haze phenomenon (Garrett and Zhao, 2006; Mitchell, 1957; Zhao and Garrett, 2015). On one hand, anthropogenic aerosol from low and middle latitudes can disturb the Arctic atmosphere, especially from Eurasia. On the other hand, stable atmospheric status with less precipitation occurs in the Arctic winter, which makes it difficult for aerosols to be removed by wet deposition (Garrett et al., 2010; Heintzenberg, 1989). As shown in Figure S1, the Arctic region has a smaller monthly average convective available potential energy (CAPE) in winter half-year, while the monthly average wind speed at 10 m above the surface is higher. Among the three study regions, the AOD of the Antarctic is slightly higher than that of the Arctic in the southern hemisphere wintertime, while the AOD of the Antarctic is the lowest in other months. The slightly higher AOD shown in the Antarctic in spring and winter compared to the other two seasons may be due to a similar reason as in the Arctic: stable atmospheric conditions and less precipitation make the aerosols difficult to be removed in spring and winter. Meanwhile, the standard deviation of AOD is also calculated

and shown as error bar in Figure 3. It can be seen that the standard deviation of AOD over the TP is larger than that over the Arctic and Antarctic, indicating that the variation of AOD over the TP is more significant.

Similar patterns of multi-year averaged seasonal variation of AOD over the three study regions were also observed using the AERONET data, which have high accuracy and are widely used in aerosol

characteristics and satellite-based AOD inversion verification studies (Holben et al., 1998; Martonchik et al., 2004; Russell et al., 2010; Yang et al., 2019). Over the TP, the multi-year averaged AOD reaches the maximum in April and the minimum in December, while the aerosol composition varies greatly at different sites (Cong et al., 2009; Pokharel et al., 2019). High AOD mainly occurs in spring, associated with the Arctic haze, and low AOD occurs in summer over the Arctic (Breider et al., 2014; Grassl and

Ritter, 2019; Rahul et al., 2014). Monthly mean values of AOD have also been investigated using the AERONET sites (Novolazarevskaya, Dome Concordia, and South Pole) over the Antarctic, which are similar to that found using CALIPSO data, with values ranging from 0.02 to 0.04 from September to March (Tomasi et al., 2015). It should be noted that due to the daytime limitation, only the AODs during the short summer period were analyzed over the Arctic and Antarctic using AERONET measurements.

**3.1.3 The long-term trend of AOD**

To study the long-term trend of AOD over the Arctic, Antarctic, and TP, the monthly AODs along with their standard deviations from June 2006 to December 2019 were calculated using valid data in the study areas. In order to remove the clear seasonal variation of AOD as found earlier in the study regions, the deseasonalized trend was carried out by calculating the AOD anomalies. The AOD anomaly here is

defined as the difference between the monthly average value of AOD in each month and the average value of AOD for that month in all years. The results of the monthly AOD anomaly over the Arctic, Antarctic, and TP are outlined in Figure 4. The solid line with red color represents the monthly AOD anomaly, the shadow region represents the single standard deviations, and the blue dotted line represents the linear trend based on deseasonalized monthly AOD anomalies from June 2006 to December 2019.

Figure 4 shows that there are no significant increasing or decreasing trends of AOD anomalies in the Arctic, Antarctic, and TP (slope = -0.00724% ~ -0. 00219%), although the linear trends show a high confidence level ($p > 0.05$). It is worth noting that the deseasonalized monthly AOD anomalies over the TP region are relatively high. There are two likely reasons. First, there are anthropogenic emission

sources over the TP region (Li et al., 2016; Zhu et al., 2019). Second, the TP is located in Central Asia

surrounded by highly polluted areas, which is easily affected by external aerosol transport (Hu et al.,

2020; Liu et al., 2015; Xia et al., 2021; Zhao et al., 2020). Figure S2 also presents the temporal variation

of seasonal average AOD from the summer of 2006 to the winter of 2019 over the TP, Arctic, and

Antarctic. As expected, AOD over the three study regions has an obvious seasonal variation trend. For

the TP, the average AOD is about 0.15 in spring, which is the most serious pollution season in the whole

year, while AOD is about 0.05 in winter, which is the cleanest season in the whole year. Boreal winter

(summer) and summer (winter) are the most polluted and cleanest seasons over the Arctic (Antarctic),

respectively. In addition, the standard deviations of seasonal AODs over the TP are between 0.0 and 0.12

due to the influence of transported aerosols from surrounding regions, which is greater than that of 0.0

to 0.05 over the Arctic and Antarctic.

**3.2    The properties of different aerosol types**

Compared with the aerosol type information from AERONET, MODIS, MISR, and OMI, aerosol types

obtained using the CALIPSO are widely used to investigate the aerosol characteristics at a local or global

scale. However, the uncertainty assessment of CALIPSO aerosol types is still a challenging task (Kahn

and Gaitley, 2015), especially for aerosol types which have similar optical properties, such as polluted

dust and smoke (Zeng et al., 2021). Aerosol subtypes from the CALIPSO V3 dataset were evaluated with

the AERONET product by previous studies, which showed the consistency of all aerosol types except

for smoke and polluted dust aerosols (Burton et al., 2013; Mielonen et al., 2009). In V4 CALIPSO aerosol

classification algorithm, several refinements were conducted to improve the accuracy of aerosol type

classification (Kim et al., 2018).

**3.2.1 Horizontal distribution**

In order to examine the spatial and temporal variability of aerosol types, the normalized annual and

seasonal averaged OFs of different aerosol types over the (a) Arctic, (b) Antarctic, and (c) TP were

presented in Figure 5 and Table 1, respectively. The number i ~ vii represent OF of clean marine, dust,

polluted continental/smoke, clean continental, polluted dust, elevated smoke, and dusty marine aerosol,

respectively.

In terms of the spatial distribution of aerosol OF among the three study regions, it can be seen from Figure 5 that the annual average OF of aerosol types is roughly similar in both the Arctic and Antarctic. The dominant aerosol type is the clean marine, followed by polluted continental/smoke and polluted dust. The annual average proportion of time with clean marine aerosol dominant in the Arctic and Antarctic is

about 32.8% and 37.5%, respectively. In contrast, the dominant aerosol types over the TP are dust type and polluted dust type, which show the dominant role for 92% time of the whole year. Figure 5 also shows that there are large differences in the spatial distribution of different aerosol types over all study regions. However, the spatial distribution of aerosol types has a distinctive feature, that is, the OFs of dust (ii), polluted continental/smoke (iii), and polluted dust (v) over the land are significantly higher than

that over the ocean area in the Arctic and Antarctic. The clean marine (i) aerosol mainly occurs in the sea area of the Arctic and Antarctic regions, and the farther away from the land, the higher the OF of clean marine aerosol. Clean continental (iv) aerosol only occurs in the land area, while dusty marine aerosol (vii) only occurs in the marine area. The OF of elevated smoke (vi) does not differ significantly between land and sea areas, which can be explained to a certain extent by the fact that the elevated smoke

aerosols in the Antarctic and Arctic are mainly transported from the outside.

There is also a significant difference in the OF spatial distribution of different aerosol types in each study region. In the Arctic, dust (ii) and polluted dust (v) aerosol has a higher frequency of occurrence over Greenland, northeastern Asia, and northern America. There are two main contributing sources. One is that the contribution of local emission (e.g. Iceland) to dust aerosols in the Arctic is significant, especially

in winter (Dagsson-Waldhauserova et al., 2019; Fan, 2013). The other is that the transport of Asian dust into the atmosphere, which was subsequently transported eastward and reached the high-latitude regions of Northern America (Tomasi et al., 2007; VanCuren et al., 2012). In contrast, polluted continental/smoke (iii) aerosol mainly occurs in Eurasia, which is mainly due to biomass burning (e.g., agricultural burning and wildfires) in the Eurasian region (Soja et al., 2006; Warneke et al., 2010). In the Antarctic, there are

obvious spatial differences in aerosol types. Specifically, dust (ii) and polluted dust (v) aerosols are the dominant aerosol types that occurred in East Antarctica. The main aerosol type in Western Antarctic (Antarctic Peninsula) is polluted continental/smoke (iii) aerosols, but there is also a certain proportion of polluted dust (v) aerosol in West Antarctica, similar to the findings reported by Li et al. (2008). The clean marine aerosol mainly occurs in the Southern Ocean and decreases drastically in the interior of Antarctica

in occurrence frequency (Teinilä et al., 2014; Virkkula et al., 2006). For the TP region, dust aerosols

occur more frequently in the north and west of the TP, which is mainly because they are close to desert

source areas, including the Tarim Basin, Qaidam Basin, and Iranian Plateau. Differently, the polluted

dust in the south of the TP has a higher frequency of occurrence, which may be due to the impact of

South Asia anthropogenic pollutants and biomass burning aerosols. Similarly, polluted

continental/smoke and elevated smoke also have a higher frequency in the southern TP.

As mentioned above, aerosol types have a distinct seasonal variation. We then investigated the seasonal

average OF of different aerosol types. In this study, the number of samples of seven aerosol types in each

study region was first counted, and then the normalized OF of different aerosol types was calculated

seasonally. Similar to the findings in Figure 5, in general, the dominated aerosol type is clean marine

over the Arctic and Antarctic. However, the normalized OF of aerosol types display a substantial seasonal

dependence (Table 1). Specifically, the proportion of clean marine aerosols is larger in the Arctic in

autumn and winter than that in spring and summer. This may be due to the near-surface wind speed in

winter half-year in the Arctic region is higher than that in summer half-year, which makes more marine

aerosols enter the atmosphere (Erickson et al., 1986; Hughes and Cassano, 2015). In the summer fire

season, the wildfires and agricultural burning occur more frequently over Siberian and North American,

which can be transported to the Arctic along with the pollutants, resulting in a high proportion of polluted

continental/smoke aerosol and elevated smoke aerosol. This notion is also supported by previous studies

(Stohl et al., 2006; Schmeisser et al., 2018; Tomasi et al., 2007). In spring, meanwhile, the proportion of

dust and polluted dust increases significantly in the Arctic, which is due to the transported dust from

Asian desert sources (Barrie, 1995). Similar results about the seasonal variation of aerosol type over the

Arctic also simulated by the GEOS-Chem model (AboEl-Fetouh et al., 2020). Different from the Arctic,

clean marine aerosol was the dominant aerosol type in the Antarctic, especially in summer, accounting

for about 61.2 %. Similar results were reported by Quinn et al. (1998). Meanwhile, there is a high

proportion of dust aerosols in the Antarctic except in winter. It is also found that polluted

continental/smoke aerosol in the southern hemisphere in winter and spring has a relatively large

proportion, consistent well with previous findings that there is more equivalent back carbon

concentrations in spring and winter than that in summer and autumn (Bodhaine, 1995; Weller et al.,

2013). Compared with the Antarctic and Arctic regions, the types of aerosols in the TP are relatively

simple, which are mainly the dust aerosol and polluted dust aerosol. In spring and summer, the proportion

of dust aerosol is relatively high, because the dust aerosols originating from the Taklimakan Desert are

transported to the internal TP through the northwesterly wind under the topographic blocking (Liu et al.,

2015; Jia et al., 2015). In autumn and winter, the emission of anthropogenic aerosol increases, resulting

in higher OF of polluted continental/smoke; and elevated smoke aerosols also increase due to the increase

of biomass combustion (Carter et al., 2016; Cheng et al., 2020). Similar results were also found by

measuring the concentration of polycyclic aromatic hydrocarbons (PHAs) in soil, which is the by-

products of incomplete combustion of organic matter (Tao et al., 2011). Correspondingly, the proportion

of polluted dust aerosol, which is the mixture of anthropogenic aerosol and dust aerosol, increases in

autumn and winter over the TP.

### 3.2.2 The vertical extinction coefficient of dominant aerosol type

Knowledge of aerosol extinction coefficient is necessary to enhance our understanding of how

atmospheric aerosols impact the weather and climate to a certain extent (Jung et al., 2019). The extinction

properties of the three typical aerosol types (including dust, elevated smoke, and polluted dust) and the

average value of the extinction of all aerosol types were retrieved in the CALIPSO L3 aerosol profile

product. In this study, the seasonal average aerosol extinction coefficient profiles (Spring: (a) ~ (c);

Summer: (d) ~ (f); Autumn: (g) ~ (i); Winter: (j) ~ (l)) over the Arctic, Antarctic, and TP were calculated

statistically and shown in Figure 6.

As Figure 6 shows, there is no doubt that the aerosol extinction coefficient profile has a significant

regional difference. In general, the aerosol extinction coefficient in the Arctic has a broad vertical

distribution at heights ranging from 0 to 12 km, but the vertical distribution of the Antarctic aerosol

extinction coefficient is uneven. In the Antarctic, the extinction layer can reach a maximum height at 11

km in winter (k) and spring (b), while it is mainly distributed below 5 km in summer (e) and autumn (h).

The vertical distribution of aerosols over the TP is more concentrated, with most aerosols distributed

between 2 and 8 km. The vertical distribution of extinction coefficients of different aerosol types also

demonstrates large regional differences. The elevated smoke in the Arctic has a larger extinction

coefficient when the altitude is greater than 2 km, especially in summer (d) and autumn (g); while in the

near-ground area (altitude < 2 km), dust and polluted dust have a larger extinction coefficient, which is

in good agreement with previous studies (Di Biagio et al., 2018). The extinction coefficients of aerosols

in the Antarctic have obvious seasonal characteristics. The vertical distribution patterns of extinction coefficients for the three aerosol types in spring (b) and autumn (h) are basically the same, and the extinction layers are mainly concentrated at heights below 5 km. In summer (e), the vertical distributions of extinction coefficients are quite different among the different types of aerosols. The elevated smoke is mainly concentrated at heights about 3 km, while the dust-related aerosol types are more distributed at heights below 2 km. In winter (k), on the contrary, the extinction coefficient of dust and elevated smoke increases significantly above 5 km, and the polluted dust aerosols have large extinction coefficients under 5 km. Unlike the Arctic and Antarctic regions, the extinction coefficients of smoke and dust-related aerosols over the TP region are larger at heights of 4 - 9 km and 2 - 9 km, respectively. From the perspective of seasonal variation, the extinction coefficient of dust aerosol is larger in spring (c) and summer (f) than in autumn (i) and winter (l). In contrast, the extinction coefficient profile of polluted dust aerosol shows larger values in spring (c) and autumn (i). These vertical distribution information of aerosol can help better understand the sources and impacts of aerosols over the three study regions in future. For example, aerosol information below clouds could be particularly important for aerosol-cloud interaction study. Note that the vertical distribution of aerosol characteristics could also be influenced by the topography in each region, which is out of the scope of current study.

### 3.2.3 Vertical distribution

Aerosol types not only have significant spatial and temporal variations, but also vary with height. CALIPSO data provides the vertical distribution of aerosol types at 208 levels, ranging from surface to 12 km. We here investigate the vertical distribution of seven aerosol types, as shown in Figure 7. The results show that most of the aerosol types in the Arctic and Antarctic regions have similar vertical distribution patterns, except for dust and polluted dust. Clean marine, polluted continental/smoke, clean continental, and dusty marine mainly occur near the surface with altitudes below 3 km. Different from the above four types of aerosols, the elevated smoke is found more at higher altitudes extending up to 8 km and 4 km with the highest OF at about 2.5 km in the Arctic and Antarctic regions, respectively, which indicates that the main source of elevated smoke is external transport. In addition, polluted continental/smoke aerosols occur more frequently in the Arctic region than in the Antarctic region. This is mainly due to the fact that the Arctic region is surrounded by more continents and more continental pollutants can enter the Arctic region. Compared with the Arctic, the dust and polluted dust in the

Antarctic region have obvious vertical distribution characteristics. The dust and polluted dust aerosols are mainly located within 3 - 5 km in the Antarctic, which indicates that the dust-related aerosols in the Antarctic area are mainly transported from outside through the upper air (Li et al., 2008). Similar to previous studies, dust-related aerosol layers over the TP appear most frequently at approximately 4 – 7/8 km above the mean sea level, where the plumes likely originate from the nearby Taklimakan Desert (Huang et al., 2007; Liu et al., 2015; 2020b; Xu et al., 2020).

## 3.3 Back trajectory

In order to better understand the origins of the air masses arriving in the study regions, the latest version (V5.0.0) of the HYSPLIT model was used in this study to simulate the back trajectories of air masses. Eleven sites listed in Table S1 were selected in this study, and the 14-day back trajectories for the Arctic, Antarctic, and TP sites were simulated. A total of 3,432 ($11\times2\times12\times13$: 11 sites, 2 times per month, 12 months per year, and a total of 13 years from 2007 to 2019) back trajectories were computed at a height of 500 m above the surface at all eleven sites. The seasonal climatologies (January 2007 to December 2019) of air mass trajectories were created and the cluster analysis was implemented to examine the long-range transport pathways of air masses. The cluster analysis determines the final number of clusters based on the total spatial variance (Draxler and Hess, 1998). Figure 8 reveals the seasonal climatological characteristics of the back trajectories after cluster analysis. It can be seen that the back trajectories over different study regions have distinctive characteristics, especially in the TP region. It is worth noting that due to the fact that coarse resolution reanalysis data is difficult to describe meteorological fields under complex terrain conditions, the back trajectories of air masses simulated by HYSPLIT may have a large error over the TP region. Compared with the Antarctic, the air mass trajectory in the Arctic region has a shorter transport distance. This is most likely due to the fact that the temperature in the Arctic is higher than that in the Antarctic, which decreases the pressure gradient and reduces the near-surface wind speed. In the Arctic, the difference of back trajectories between summer and winter half-year is obvious, with more proportion of air masses from the Eurasian in winter and spring. At the same time, Asian dust storms prevail in spring, resulting in a greater proportion of dust and polluted dust in spring. In contrast, the influence of external transport of aerosols is relatively small in autumn, and the larger near-surface wind speed allows more marine aerosols to enter the atmosphere, which together make the contribution of clean marine aerosols in autumn relatively large in the Arctic.

In the Antarctic region, the seasonal difference in air mass trajectories is relatively small compared with the Arctic region, and the air mass trajectories were mainly controlled by circumpolar westerly winds (Ravi et al., 2011). While it is not clearly shown by the air mass back trajectory simulation results, dust and polluted dust over East Antarctica was likely caused by the transport from South America and Africa,

and the polluted dust over West Antarctica was more likely affected by the aerosol transport from South America and Australia. Generally speaking, under the influence of steady and strong westerly winds, dust and carbonaceous aerosols in South America, Australia, and Africa have a certain impact on Antarctic pollution (Li et al., 2008; McConnell et al., 2007; Zou et al., 2018).

Different from the Arctic and Antarctic, the back trajectories of air masses over the TP have significant

seasonal variation. In spring and summer, the air masses located on the northern slope of the TP mainly come from the northern desert area. In autumn, the air masses from the north begin to weaken, while the air masses from Iranian Plateau begin to increase and reach the maximum in winter (93.15 %). For the site on the southern slope of the TP, the air masses mainly come from the Iranian Plateau in spring and winter, while in summer they mainly come from South Asia, which makes the TP more vulnerable to

pollution from the Indian Peninsula and South Asia. Similar to the site on the northern slope of the TP, the back trajectories of air masses at the eastern slope site are greatly affected by the Tarim Basin and Qaidam Basin in spring, but are mainly affected by the Iranian Plateau and the western part of the TP in autumn and winter. Differently, in summer, the back trajectories of air masses are not only from the Tarim Basin and Qaidam Basin, but also from the southern part of the TP with about 27.54% of air

masses.

## 4    Summary and Conclusions

Aerosols play a crucial role in the radiative budget of the Earth-atmosphere system, but due to insufficient understanding of aerosol properties, at least partly, the uncertainty of the total radiative forcing by aerosols in the climate mode is still the largest. Understanding the properties of aerosols is highly

demanded. The satellite active remote sensing can make up for the insufficiency of ground-based remote sensing to obtain long-term and large-scale aerosol properties. In this study, the spatial and temporal distribution of the aerosol optical depth (AOD) and aerosol type over the Arctic, Antarctic, and Tibetan Plateau (TP) regions were investigated. In addition, eleven typical sites were selected and the back

trajectories of air masses were simulated using the Hybrid Single-Particle Lagrangian Integrated Trajectory (HYSPLIT) model. The main findings are as follows.

The distribution of AOD over the three study regions shows distinctive spatial and seasonal differences. In general, the AOD over the Arctic and Antarctic decreases with the increasing latitude. In the Arctic, the AOD over land is greater than that over the ocean, while the opposite is true for the Antarctic. Eurasia and the Ross Sea are the high AOD areas in the Arctic and Antarctic, respectively. The annual average AOD over the TP region (0.098) is about twice that of the Arctic (0.046) and four times that of the Antarctic (0.024). The seasonal variation of AOD over the TP is the most distinctive due to the influence of transported aerosols from surrounding high emission regions. The maximum AOD occurs in spring and summer over the TP, while occurs from late autumn to early spring in the Arctic and in winter and spring in the Antarctic.

The deseasonalized trend of AOD (called AOD anomaly) over the three regions was also investigated. The result shows that there were no obvious temporal trends in the AOD anomalies over the Arctic, Antarctic, and TP. Compared with the Antarctic and Arctic, the AOD anomalies over the TP have obvious fluctuations, which indicates that the TP is more susceptible to the influence of highly varied aerosols from different regions. In the Arctic, the aerosol extinction coefficient has a broad vertical distribution at heights from the surface to 12 km. Moreover, the extinction coefficient of elevated smoke and polluted dust in the upper layer is large in the Arctic, especially in summer and autumn. In the Antarctic, the vertical distribution of aerosol extinction has obvious seasonal differences. Dust aerosol has a large extinction coefficient at heights 5 - 11 km in winter, while in other seasons, the aerosol extinction coefficient is large at heights below 5 km.

The multi-year average (June 2006 - December 2019) occurrence frequency (OF) of aerosol types was also examined. The OF of different aerosol types demonstrates significant spatial differences. In the Antarctic and Arctic regions, the dominant aerosol type is the clean marine type, followed by polluted continental/smoke and polluted dust aerosol types. Clean marine aerosol types are mainly distributed over the seas of the polar regions, and polluted continental/smoke and polluted dust are mainly distributed over the land regions. In the Arctic, polluted continental/smoke aerosol types are mainly distributed in the northern part of Europe, while polluted dust aerosols are widely distributed in the northern parts of Asia and America along with the Greenland Island region. In the Antarctic, dust and polluted dust aerosol

types are mainly distributed in East Antarctica, and polluted continental/smoke aerosol types are mainly distributed in the Antarctic Peninsula. In the TP region, the main aerosol types in the north and south of

the TP are dust and polluted dust, respectively. The normalized seasonal OF of seven aerosol types is further investigated. The result shows that the OF of each aerosol type in different regions has obvious seasonal variations. Regarding the vertical distribution of the OF of aerosol types, dust, polluted dust, and elevated smoke have a relatively large OF at higher altitudes. And the maximum altitude with a noticeable OF of these types of aerosols is higher in the Antarctic than in the Arctic. Different from the

Arctic and Antarctic, the dust-related aerosol layers over the TP appear most frequently at heights approximately 4 - 7 km above the mean sea level.

The back trajectories of air masses indicate that the Arctic region is vulnerable to mid-latitude pollutants, especially in winter and spring, while the Antarctic region is less affected by the mid-latitude pollutants. Different from that in the Arctic and Antarctic, the air mass trajectories over the TP have obvious seasonal

variations.

**Data Availability.**

The CALIPSO dataset were obtained from https://earthdata.nasa.gov/. Surface elevation data from Shuttle Radar Topography Mission (SRTM) were downloaded from http://srtm.csi.cgiar.org/. HYSPLIT data are provided by the NOAA READY website (http://www.ready.noaa.gov).

**Acknowledgement.**

This research was supported by the Strategic Priority Research Program of the Chinese Academy of Sciences (Grant number XDA19070202), the Natural Science Foundation of China (91837204, 41925022). The authors gratefully acknowledge the data support from NASA for making CALIPSO L3 datasets accessible in public. We also gratefully thank the NOAA Air Resources Laboratory (ARL) for

the provision of the HYSPLIT transport and dispersion model. We thank the reviewers of this paper for their valuable comments which helped improve the manuscript.

**Author contributions.**

CFZ designed the research, and CFZ and YKY carried out the research and wrote the manuscript. QW and XCY contributed to run the HYSPLIT model. ZYC provided constructive comments and revised the

manuscript many times. HF provided constructive comments on this research. All authors made

substantial contributions to this work.

**Competing interests.**

The authors declare that they have no conflict of interest.

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

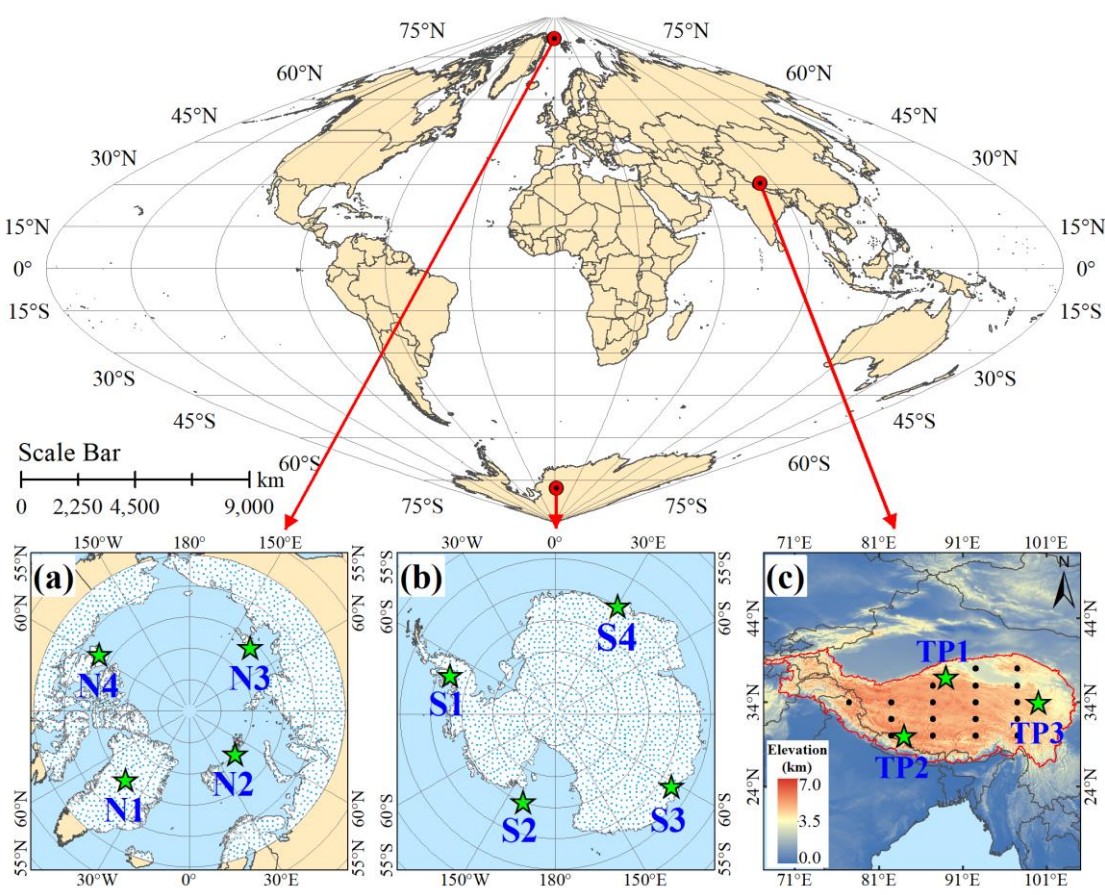

Figure 1: Geographical map of the three study areas, (a) Arctic, (b) Antarctic, and (c) TP. Among them, the
white background and blue pints represent the land within the study area. In (c), the black dots represent the
center of the TP inner pixel corresponding to CALIPSO L3 aerosol data, the green pentagrams represent the
site of aerosol back trajectory study, the red line represents the boundary of the TP, and the color represents
the surface elevation data from Shuttle Radar Topography Mission (SRTM) on http://srtm.csi.cgiar.org/.

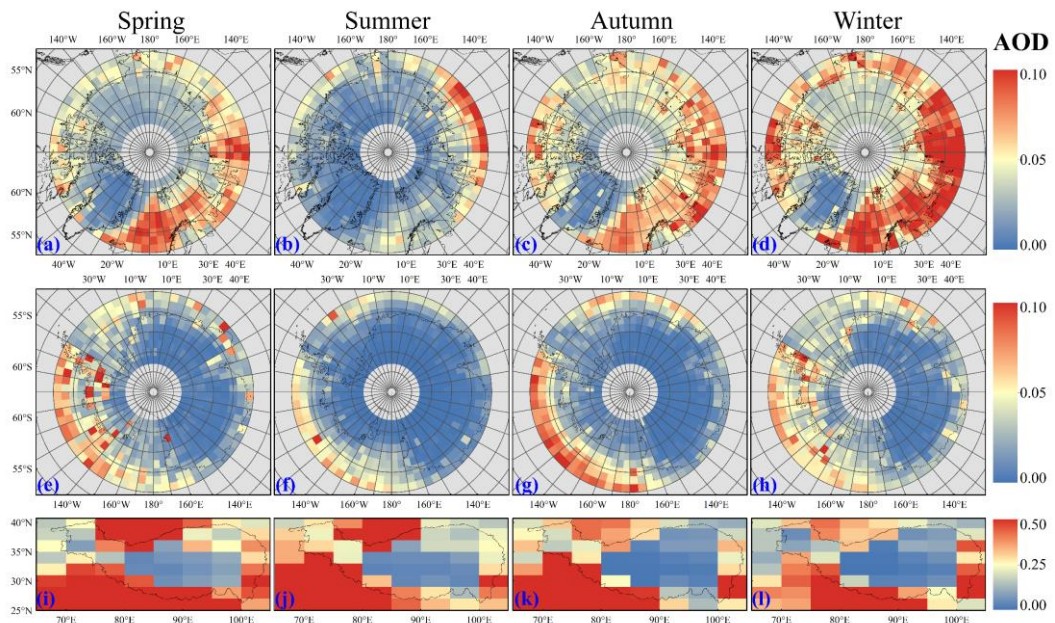

Figure 2: Seasonal averaged AOD distribution for thirteen years (June 2006 to December 2019) over the
Arctic, Antarctic, and TP. Four columns represent four seasons. (a) ~ (d), (e) ~ (h), and (i) ~ (l) represent the
spatial distribution of aerosols in the Arctic, Antarctic, and TP, respectively.

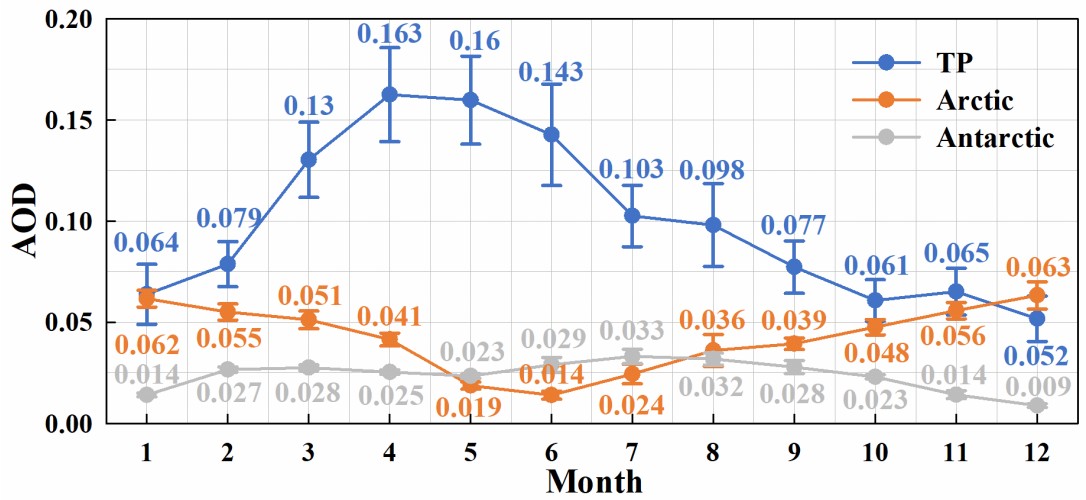

Figure 3: The monthly averages (dots) and standard deviations (bars) of AODs for the study period from June
2006 to December 2019 over the Arctic, Antarctic, and TP.

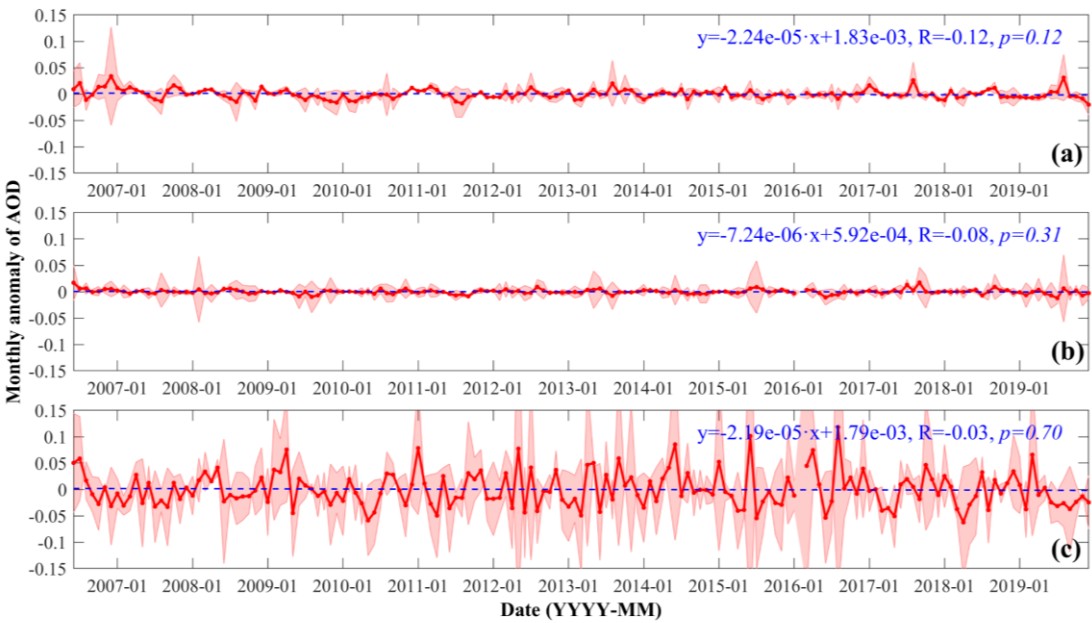

**Figure 4: Temporal variation of monthly AOD anomalies from June 2006 to December 2019 over the (a) Arctic, (b) Antarctic, and (c) TP. The red solid lines and shadows represent the deseasonalized monthly AOD anomalies and standard deviations, respectively, while the blue dotted lines represent the linear trends.**

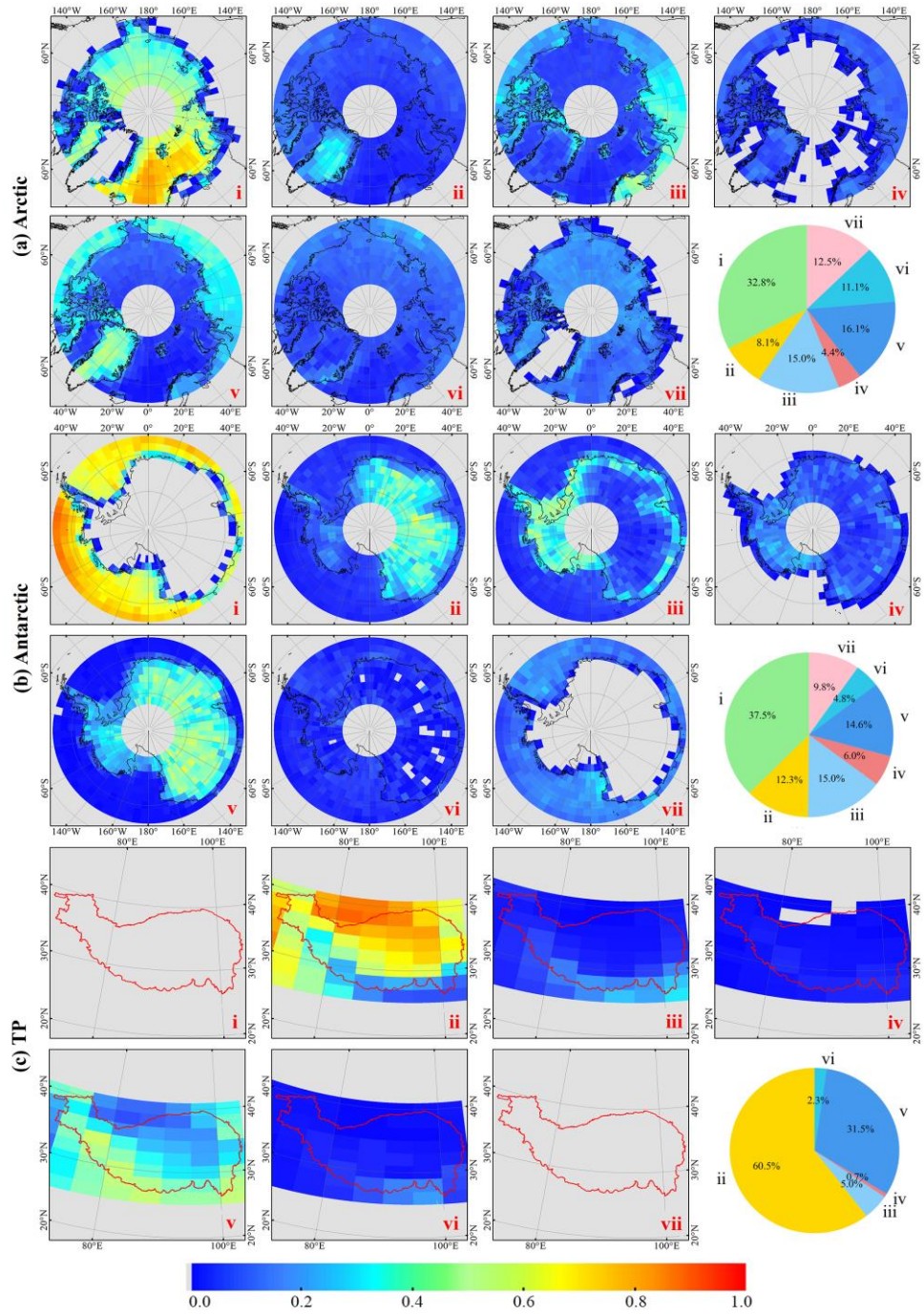

985

**Figure 5: The annual averaged OF maps during the study period from June 2006 to December 2019 for seven aerosol types defined by CALIOP products over the (a) Arctic, (b) Antarctic, and (c) TP. The number i ~ vii represent clean marine, dust, polluted continental/smoke, clean continental, polluted dust, elevated smoke, and dusty marine, respectively. The pie represents the annual average OF of all pixels for seven aerosol types.**

990

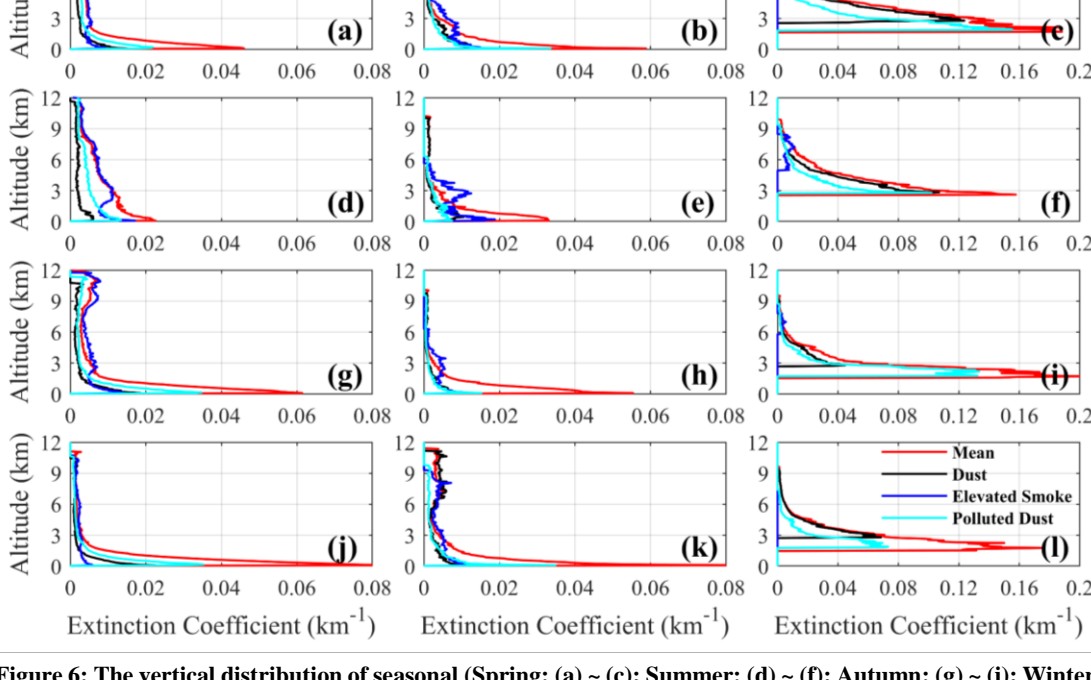

**Figure 6: The vertical distribution of seasonal (Spring: (a) ~ (c); Summer: (d) ~ (f); Autumn: (g) ~ (i); Winter: (j) ~ (l)) averaged aerosol extinction coefficient at 532 nm during the study period from June 2006 to December 2019 over the Arctic (left panel), Antarctic (middle panel), and TP (right panel), including the mean extinction coefficient (red solid line), dust extinction coefficient (black solid line), Elevated smoke extinction coefficient (blue solid line), and polluted dust extinction coefficient (cyan solid line) over the Arctic (a, d, g, and j), Antarctic (b, e, h, and k), and TP (c, f, i, and l).**

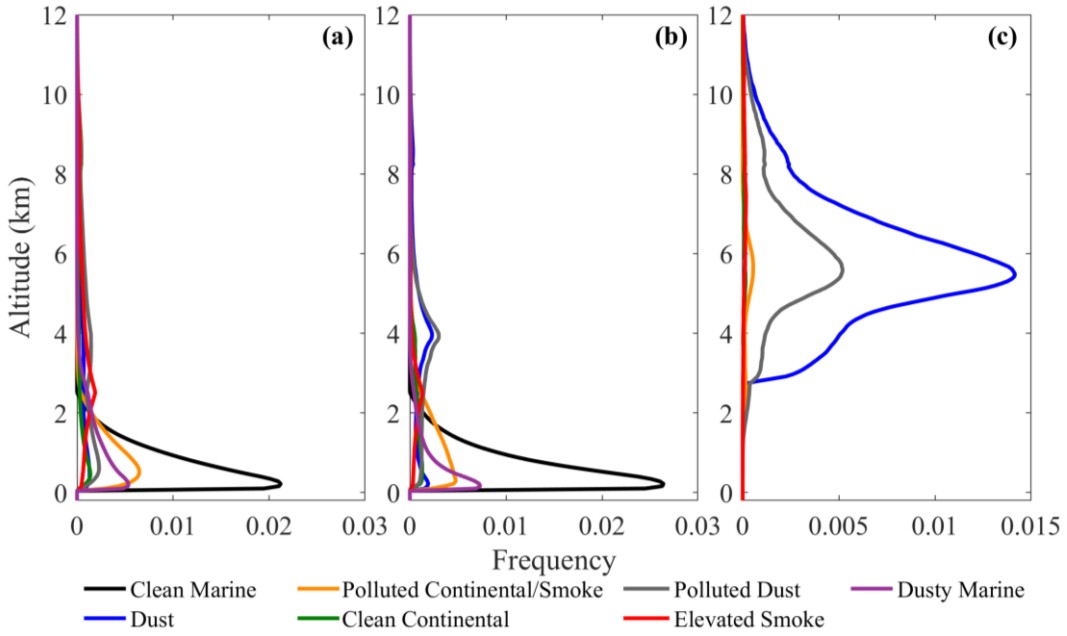

**Figure 7: The vertical distribution of multi-year (June 2006 to December 2019) average OF of aerosol types over the (a) Arctic, (b) Antarctic, and (c) TP.**

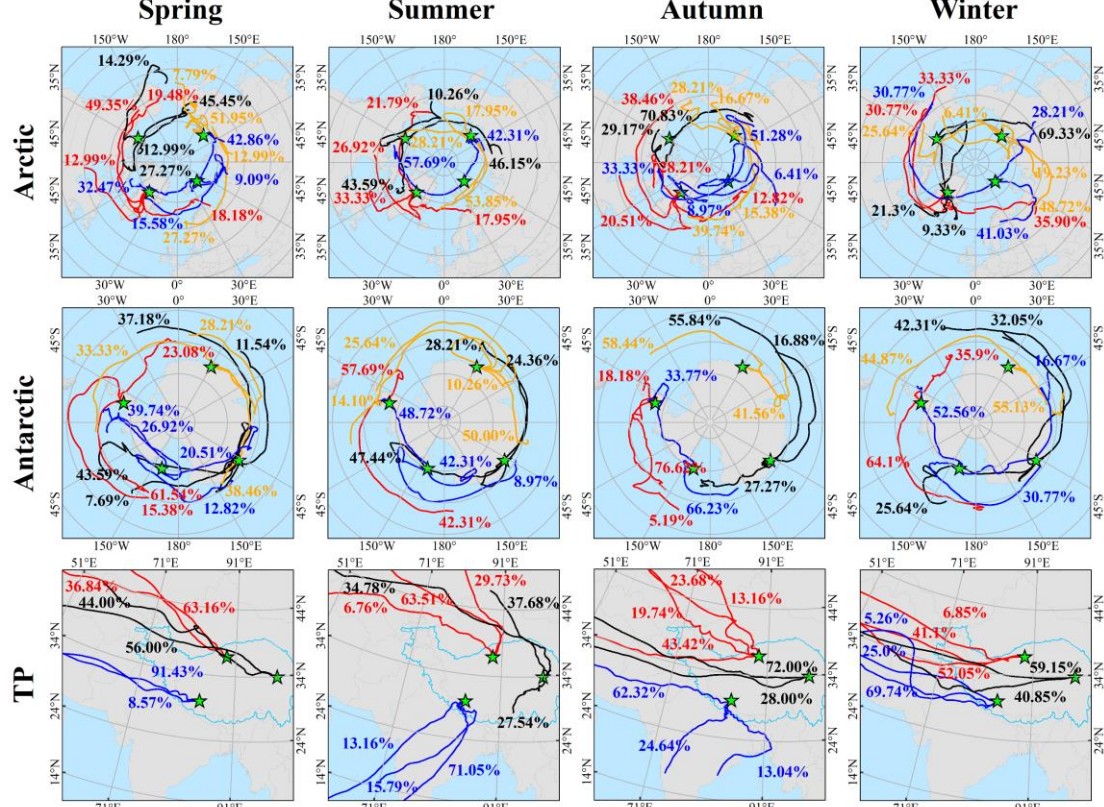

**Figure 8: The seasonal average characteristics of the back trajectories for the study period from January 2007 to December 2019 at each selected site over the Arctic, Antarctic, and TP.**

**Table 1: The normalized seasonal average OF of seven aerosol types during the study period from June 2006 to December 2019 over the Arctic, Antarctic, and TP.**

| Region | Season | Aerosol Types | | | | | | |
|--------|--------|------|------|------|------|------|------|------|
| | | Clean marine | Dust | Polluted continental /smoke | Clean continental | Polluted dust | Elevated smoke | Dusty marine |
| Arctic | Spring | 0.244 | 0.113 | 0.119 | 0.045 | 0.231 | 0.125 | 0.123 |
| | Summer | 0.131 | 0.041 | 0.192 | 0.051 | 0.184 | 0.342 | 0.059 |
| | Autumn | 0.418 | 0.067 | 0.143 | 0.049 | 0.124 | 0.107 | 0.092 |
| | Winter | 0.371 | 0.073 | 0.163 | 0.041 | 0.134 | 0.069 | 0.149 |
| Antarctic | Spring | 0.344 | 0.143 | 0.154 | 0.054 | 0.137 | 0.052 | 0.116 |
| | Summer | 0.612 | 0.137 | 0.082 | 0.007 | 0.052 | 0.021 | 0.089 |
| | Autumn | 0.34 | 0.139 | 0.144 | 0.069 | 0.200 | 0.043 | 0.065 |
| | Winter | 0.359 | 0.096 | 0.169 | 0.070 | 0.133 | 0.056 | 0.117 |
| TP | Spring | 0.000 | 0.731 | 0.015 | 0.004 | 0.238 | 0.012 | 0.000 |
| | Summer | 0.000 | 0.714 | 0.021 | 0.005 | 0.244 | 0.016 | 0.000 |
| | Autumn | 0.000 | 0.554 | 0.053 | 0.008 | 0.361 | 0.024 | 0.000 |
| | Winter | 0.000 | 0.331 | 0.144 | 0.014 | 0.463 | 0.048 | 0.000 |

1005