# Peer review of "Aerosol Characteristics in the Three Poles of the Earth as Characterized by CALIPSO"

_Atmospheric Chemistry and Physics, 2020_

## Referee Comment (RC1) · Anonymous Referee #1 · 2 Dec 2020

Using CALIOP data during 2006-2019, the authors presented a detailed picture of aerosol vertical profiles in three key background regions, i.e., the Arctic, Antarctic, and Tibetan Plateau. Given aerosol's important role in climate research and we are not fully clear how aerosol varies, especially with regard to the vertical profile, this work fills a big gap in this regard. Therefore, I suggest to accept this submission after following issues are addressed. 1. Suggest to add extra analyses based on AERONET data to support CALIOP results or to cite published results to support the results presented here, for example, the seasonal variations. 2. Pls add the standard deviation of monthly mean in Fig. 3. 3. Pls add some words on how to distinguish aerosol types by CALIOP measurements and uncertainty associated with this method. My understanding is that it is very hard to separate polluted dust from smoke, if so, some words discussing un-

certainty about occurrence of some specific aerosol types should be added. 4. Could you pls present some results on seasonal trend of AOD since long-range transport of external sources to these three regions is seasonal dependent. 5. L35-40, suggest to change to "depends on aerosol charateristics and underlying surface 6. L50, references not suitable, suggest add original AERONET references, for instance, Dubovik and King, 2000; Dubovik et al., 2006 7. L58, it looks strange because sentence before talk AERONET, but after that, you compare passive and active satellite remote sensing, there is no words on passive satellite remote sensing. 8. L97, discussion of dust transport to the TP is originally discussed by Huang et al., 2007 and further supported by Xia et al. (2008), with regarding long-range transport from south Asia, some references should be added including Xia et al. (2011); Lu et al. (2012); Zhao et al. (2015). A detailed discussion on this issue can be found in a overview paper (Xia et al., 2021, AR). 9. L182, pls add standard deviation to the mean value 10 L190-195, it seems transport of dust to the TP mainly occurs in summer, on the other hand, transport of fine particles from South Asia mainly occurs in dry season.
* * *

---

## Referee Comment (RC2) · Anonymous Referee #2 · 3 Dec 2020

General comments: The manuscript entitled "Aerosol Characteristics in the Three Poles of the Earth Observed by CALIPSO" focuses on aerosol characteristics of the Arctic, Antarctic, and Tibetan Plateau by using CALIPSO L3 data and HYSPLIT model. The results show that the AODs over three regions have obvious spatial and temporal feature. Different type of aerosol has remarkable spatial and seasonal. Overall, this manuscript is clear and well written. Some concerns are needed to address.

1. Page 1, Line 18-19, the sentence should be "The annual average AODs over the Arctic, Antarctic, and TP are 0.046......", or "The annual mean values of AOD over the Arctic, Antarctic, and TP are 0.046......". 2. What is the basis of selecting the simulation points of backward trajectory? I recommend authors to add a simulation point over the eastern TP? 3. Page 7, Line 178, the authors described "decrease with the

increase of latitude in any season", however, Figure 2e shows some areas with high AODs at high latitudes of Antarctic in spring. 4. Page 7, Line 190, "high aerosol concentrations in the Arctic and Antarctic mainly occur in winter and spring", The aerosol concentration of Arctic should be higher in late fall to early spring, not just winter and spring (Figure 3). 5. Page 7, Line 174, "the AOD averaged between Jun 2006 and Dec 2019"; Page 7, Line 187, "the monthly variations of multi-year (June 2016 - December 2019) average AODs"; Page 8, Lines 207-208, "the monthly AODs along with their standard deviations from June 2007 to December 2019". Why not use the AOD data for same period? 6. Page 8, Lines 219-221, "First, there are anthropogenic emission sources over the TP region. Second, the TP is located in Central Asia surrounded by highly polluted areas, which is easily affected by external aerosol transport". The corresponding evidence or reference is needed. 7. Page 8, Lines 219-221, "the wind speed in winter half-year in the Arctic region is higher than that in summer half-year". However, in Lines 199-200, "On the other hand, stable atmospheric status with less precipitation occurs in the Arctic winter". What is the real situation? It needs a verified evidence from satellite or reanalysis data. 8. Page 11, Line 287, "due to the northerly jet over the TP carrying dust aerosols to the internal TP". Generally, there is westerly jet over TP. Please explain the existence of "the northerly jet over the TP. 9. Page 11, Lines 287-288, "In autumn and winter, the emission of anthropogenic aerosol increases". Lines 290-291, "the increase of biomass combustion". Some evidences or references are needed . 10. Page 11, Lines 309-310, "while in the near-ground area (altitude < 2 km), dust and polluted dust have a larger extinction coefficient". However, there is a high value of the dust extinction coefficient above 9 km in Arctic in winter (Figure 6j), what is the possible reason? 11. Page 12, Lines 316-318, "the extinction coefficient of dust and elevated smoke increases significantly above 5 km, and the polluted dust aerosols have large extinction coefficients under 5 km". Why do the dust and polluted dust occur at different altitudes? 12. Page 12, Lines 320-321, "From the perspective of seasonal variation, the vertical distribution of dust-related aerosol extinction coefficient is larger in spring (c) and summer (f) than in autumn (i) and winter

(l)". However, as shown in Fgiure 6i, the polluted dust extinction coefficient at 3 km is significantly higher in autumn than in spring and summer. 13. Page 12, Lines 331-333, "Different from the above aerosol types, the highest OF of elevated smoke occurs at a height of about 2.5 km, and occurs at 8 km and 4 km, respectively in the Arctic and Antarctic regions." The meaning is unclear. 14. Page 11, Line 292, Page 12, Line 325, has the effect of topography in each region been considered when analyzing the vertical distribution of various aerosols? For example, the average altitude of TP is 4 km, while in Antarctic is 2.3 km. 15. Page 13, Line 347, "(10 × 2 × 12 × 13)". What do they represent respectively? 16. Page 14, Lines 371-377, from Figure 5c,âĚś, it can be found that there are more dust aerosols over north and northeast TP, however, the backward trajectory does not catch the contribution of the dust from Qaidam Basin.

---

## Author Comment (AC1) · 21 Feb 2021

We thank the reviewers for their thoughtful, valuable, and detailed comments and suggestions that have helped us improve the paper quality. Our detailed responses (Blue) to the reviewers' questions and comments (*Italic*) are listed below.

Reviewer 1:

*Using CALIOP data during 2006-2019, the authors presented a detailed picture of aerosol vertical profiles in three key background regions, i.e., the Arctic, Antarctic, and Tibetan Plateau. Given aerosol's important role in climate research and we are not fully clear how aerosol varies, especially with regard to the vertical profile, this work fills a big gap in this regard. Therefore, I suggest to accept this submission after following issues are addressed.*
We highly appreciate the positive evaluation about our study.

1. *Suggest to add extra analyses based on AERONET data to support CALIOP results or to cite published results to support the results presented here, for example, the seasonal variations.*

   We agree with the reviewer and more descriptions about the studies of aerosol properties based on AERONET measurements were added with citations at Lines 229-240: **"Similar patterns of multi-year averaged seasonal variation of AOD over the three study regions were also observed using the AERONET data, which have high accuracy and are widely used in aerosol characteristics and satellite-based AOD inversion verification studies (Holben et al., 1998; Martonchik et al., 2004; Russell et al., 2010; Yang et al., 2019). Over the TP, the multi-year averaged AOD reaches the maximum in April and the minimum in December, while the aerosol composition varies greatly among different sites (Cong et al., 2009; Pokharel et al., 2019). High AOD mainly occurs in spring associated with the Arctic haze, and low AOD occurs in summer over the Arctic (Breider et al., 2014; Grassl and Ritter, 2019; Rahul et al., 2014). Monthly mean values of AOD have also been investigated using the AERONET sites (Novolazarevskaya, Dome Concordia, and South Pole) over the Antarctic, which are similar to that found using CALIPSO data, with values ranging from 0.02 to 0.04 from September to March (Tomasi et al., 2015). It should be noted that due to the daytime limitation, only the AODs during the short summer period were analyzed over the Arctic and Antarctic using AERONET measurements."**

   References:

   Holben, B., Eck, T., Slutsker, I., Tanre, D., Buis, J., Setzer, A., Vermote, E., Reagan, J., Kaufman, Y., Nakajima, T., Lavenu, F., Jankowiak, I., and Smirnov, A.: AERONET - A federated instrument network and data archive for aerosol characterization, Remote Sensing of Environment, 66, 1-16, 10.1016/s0034-4257(98)00031-5, 1998.

   Martonchik, J., Diner, D., Kahn, R., Gaitley, B., and Holben, B.: Comparison of MISR and AERONET aerosol optical depths over desert sites, Geophysical Research Letters, 31, 10.1029/2004gl019807, 2004.

   Russell, P., Bergstrom, R., Shinozuka, Y., Clarke, A., DeCarlo, P., Jimenez, J., Livingston, J., Redemann, J., Dubovik, O., and Strawa, A.: Absorption Angstrom Exponent in AERONET and related data as an indicator of aerosol composition, Atmos. Chem. Phys., 10, 1155–1169, https://doi.org/10.5194/acp-10-1155-2010, 2010.

   Yang, Y., Zhao, C., Sun, L., and Wei, J.: Improved Aerosol Retrievals Over Complex Regions Using NPP Visible Infrared Imaging Radiometer Suite Observations, Earth and Space Science, 6,

629-645, 10.1029/2019ea000574, 2019.

Cong, Z., Kang, S., Smirnov, A., and Holben, B.: Aerosol optical properties at Nam Co, a remote site in central Tibetan Plateau, Atmospheric Research, 92, 42-48, 10.1016/j.atmosres.2008.08.005, 2009.

Pokharel, M., Guang, J., Liu, B., Kang, S., Ma, Y., Holben, B., Xia, X., Xin, J., Ram, K., Rupakheti, D., Wan, X., Wu, G., Bhattarai, H., Zhao, C., and Cong, Z.: Aerosol Properties Over Tibetan Plateau From a Decade of AERONET Measurements: Baseline, Types, and Influencing Factors, Journal of Geophysical Research-Atmospheres, 124, 13357-13374, 10.1029/2019jd031293, 2019.

Breider, T. J., Mickley, L. J., Jacob, D. J., Wang, Q., Fisher, J. A., Chang, R. Y. W., and Alexander, B.: Annual distributions and sources of Arctic aerosol components, aerosol optical depth, and aerosol absorption, Journal of Geophysical Research-Atmospheres, 119, 4107-4124, 10.1002/2013jd020996, 2014.

Grassl, S., and Ritter, C.: Properties of Arctic Aerosol Based on Sun Photometer Long-Term Measurements in Ny-angstrom lesund, Svalbard, Remote Sensing, 11, 10.3390/rs11111362, 2019.

Rahul, P., Sonbawne, S., and Devara, P.: Unusual high values of aerosol optical depth evidenced in the Arctic during summer 2011, Atmospheric Environment, 94, 606-615, 10.1016/j.atmosenv.2014.01.052, 2014.

Tomasi, C., Kokhanovsky, A., Lupi, A., Ritter, C., Smirnov, A., O'Neill, N., Stone, R., Holben, B., Nyeki, S., Wehrli, C., Stohl, A., Mazzola, M., Lanconelli, C., Vitale, V., Stebel, K., Aaltonen, V., de Leeuw, G., Rodriguez, E., Herber, A., Radionov, V., Zielinski, T., Petelski, T., Sakerin, S., Kabanov, D., Xue, Y., Mei, L., Istomina, L., Wagener, R., McArthur, B., Sobolewski, P., Kivi, R., Courcoux, Y., Larouche, P., Broccardo, S., and Piketh, S.: Aerosol remote sensing in polar regions, Earth-Science Reviews, 140, 108-157, 10.1016/j.earscirev.2014.11.001, 2015.

2. *Pls add the standard deviation of monthly mean in Fig. 3.*

We appreciate the suggestion and added the standard deviation of monthly AOD mean in Figure 3. In addition, we also added related description at Lines 225-228: **"Meanwhile, the standard deviation of AOD is also calculated and shown as error bar in Figure 3. It can be seen that the standard deviation of AOD over the TP is larger than that over the Arctic and Antarctic, indicating that the variation of AOD over the TP is more significant."**

[Figure]

Figure 3: The monthly averages (dots) and standard deviations (bars) of AODs (June 2006 to December 2019) over the Arctic, Antarctic, and TP.

3. *Pls add some words on how to distinguish aerosol types by CALIOP measurements and uncertainty associated with this method. My understanding is that it is very hard to separate polluted dust from smoke, if so, some words discussing uncertainty about occurrence of some specific aerosol types should be added.*

We agreed with the reviewer that it is difficult to separate polluted dust from smoke partially due to the fact that they have similar optical properties. As we know, the validation of aerosol subtypes is a challenging task for at least two reasons. First, aerosol types have high spatial and temporal variability, which makes it difficult to evaluate the satellite grid observations using the fixed ground site measurements or in-situ airborne measurements. Second, due to the different instrument errors and observation methods by various instruments (such as AERONET, CALIPSO, and MODIS), the definition of aerosol types is often quite different. All of these may lead to large deviations in the verification of aerosol types (Zeng et al., 2021).

Following the reviewer's suggestion, we added descriptions about the potential uncertainties at Lines 267-275: **"Compared with the aerosol type information from AERONET, MODIS, MISR, and OMI, aerosol types obtained using the CALIPSO are widely used to investigate the aerosol characteristics at a local or global scale. However, the uncertainty assessment of CALIPSO aerosol types is still a challenging task (Kahn and Gaitley, 2015), especially for aerosol types which have similar optical properties, such as polluted dust and smoke (Zeng et al., 2021). Aerosol subtypes from the CALIPSO V3 dataset were evaluated with the AERONET product by previous studies, which showed the consistency of all aerosol types except for smoke and polluted dust aerosols (Burton et al., 2013; Mielonen et al., 2009). In V4 CALIPSO aerosol classification algorithm, several refinements were conducted to improve the accuracy of aerosol type classification (Kim et al., 2018)."**

References:

Burton, S., Ferrare, R., Vaughan, M., Omar, A., Rogers, R., Hostetler, C., and Hair, J.: Aerosol classification from airborne HSRL and comparisons with the CALIPSO vertical feature mask, Atmospheric Measurement Techniques, 6, 1397-1412, 10.5194/amt-6-1397-2013, 2013.

Kahn, R., and Gaitley, B.: An analysis of global aerosol type as retrieved by MISR, Journal of Geophysical Research-Atmospheres, 120, 4248-4281, 10.1002/2015jd023322, 2015.

Kim, M.-H., Omar, A. H., Tackett, J. L., Vaughan, M. A., Winker, D. M., Trepte, C. R., Hu, Y., Liu, Z., Poole, L. R., Pitts, M. C., Kar, J., and Magill, B. E.: The CALIPSO version 4 automated aerosol classification and lidar ratio selection algorithm, Atmospheric Measurement Techniques, 11, 6107-6135, 10.5194/amt-11-6107-2018, 2018.

Mielonen, T., Arola, A., Komppula, M., Kukkonen, J., Koskinen, J., de Leeuw, G., and Lehtinen, K.: Comparison of CALIOP level 2 aerosol subtypes to aerosol types derived from AERONET inversion data, Geophysical Research Letters, 36, L18804, 10.1029/2009gl039609, 2009.

Zeng, S., Omar, A., Vaughan, M., Ortiz, M., Trepte, C., Tackett, J., Yagle, J., Lucker, P., Hu, Y., Winker, D., Rodier, S., and Getzewich, B.: Identifying Aerosol Subtypes from CALIPSO Lidar Profiles Using Deep Machine Learning, Atmosphere, 12, 10. 10.3390/atmos12010010, 2021.

4. *Could you pls present some results on seasonal trend of AOD since long-range transport of external sources to these three regions is seasonal dependent.*

Thank the reviewer for the valuable suggestion. We have added some results on the seasonal trend of AOD in Lines 257-265: "**Figure S2 also presents the temporal variation of seasonal average AOD from the summer of 2006 to the winter of 2019 over the TP, Arctic, and Antarctic. As expected, AOD over the three study regions has an obvious seasonal variation trend. For the TP, the average AOD is about 0.15 in spring, which is the most serious pollution season in the whole year, while AOD is about 0.05 in winter, which is the cleanest season in the whole year. Boreal winter (summer) and summer (winter) are the most polluted and cleanest seasons over the Arctic (Antarctic), respectively. In addition, the standard deviations of seasonal AODs over the TP are between 0.0 and 0.12 due to the influence of transported aerosols from surrounding regions, which is greater than that of 0.0 to 0.05 over the Arctic and Antarctic.**"

[Figure]

Figure S2: Temporal variation of seasonal average AOD from the summer of 2006 to the winter of 2019 over the TP (red), Arctic (green), and Antarctic (blue). The shaded area represents the standard deviation.

5. *L35-40, suggest to change to "depends on aerosol characteristics and underlying surface.* Modified.

6. *L50, references not suitable, suggest add original AERONET references, for instance, Dubovik and King, 2000; Dubovik et al., 2006.*
   Thanks for helping figure this out. We changed it.

7. *L58, it looks strange because sentence before talk AERONET, but after that, you compare passive and active satellite remote sensing, there is no words on passive satellite remote sensing.*
   We appreciate the comment and agree with the reviewer. We have added descriptions about passive satellite remote sensing at Lines 58-69: "**Passive satellite remote sensing also can be used to obtain aerosol properties. In general, passive remote sensing can only obtain two-dimensional aerosol characteristics, but cannot obtain aerosol vertical structure information. Several AOD retrieval algorithms based on passive remote sensing have been developed over the past decade, such as Dark Target (DT), Dark Water, Deep Blue (DB), and Multi-Angle Implementation of Atmospheric Correction (MAIAC), structure-function algorithm, and so on (Hsu et al., 2013; Hsu**

et al., 2004; Kaufman et al., 1997; Levy et al., 2013; Lyapustin et al., 2018; Martonchik et al., 1998; Tanre et al., 1988). In terms of aerosol type, Multi-angle Imaging SpectroRadiometer (MISR) instrument, which has nine view angles along the flight path (Diner et al., 1998), is sensitive to the size and shape of aerosols (Diner et al., 2008). Ozone Monitoring Instrument (OMI) includes ultraviolet bands, which can be used to retrieve aerosol optical parameters, such as absorbing aerosol optical depth, single scattering albedo, and aerosol index (Marey et al., 2011; Torres et al., 2007)."

References:

Hsu, N., Jeong, M., Bettenhausen, C., Sayer, A., Hansell, R., Seftor, C., Huang, J., and Tsay, S.: Enhanced Deep Blue aerosol retrieval algorithm: The second generation, Journal of Geophysical Research-Atmospheres, 118, 9296-9315, 10.1002/jgrd.50712, 2013.

Hsu, N., Tsay, S., King, M., and Herman, J.: Aerosol properties over bright-reflecting source regions, IEEE Transactions on Geoscience and Remote Sensing, 42, 557-569, 10.1109/tgrs.2004.824067, 2004.

Kaufman, Y., Tanre, D., Remer, L., Vermote, E., Chu, A., and Holben, B.: Operational remote sensing of tropospheric aerosol over land from EOS moderate resolution imaging spectroradiometer, Journal of Geophysical Research-Atmospheres, 102, 17051-17067, 10.1029/96jd03988, 1997.

Levy, R., Mattoo, S., Munchak, L., Remer, L., Sayer, A., Patadia, F., and Hsu, N.: The Collection 6 MODIS aerosol products over land and ocean, Atmospheric Measurement Techniques, 6, 2989-3034, 10.5194/amt-6-2989-2013, 2013.

Lyapustin, A., Wang, Y., Korkin, S., and Huang, D.: MODIS Collection 6 MAIAC algorithm, Atmospheric Measurement Techniques, 11, 5741-5765, 10.5194/amt-11-5741-2018, 2018.

Martonchik, J., Diner, D., Kahn, R., Ackerman, T., Verstraete, M., Pinty, B., and Gordon, H. R.: Techniques for the retrieval of aerosol properties over land and ocean using multiangle imaging, IEEE Transactions on Geoscience and Remote Sensing, 36, 1212-1227, 10.1109/36.701027, 1998.

Tanre, D., Deschamps, P. Y., Devaux, C., and Herman, M.: Estimation of Saharan aerosol optical thickness from blurring effects in thematic mapper data, Journal of Geophysical Research-Atmospheres, 93, 15955-15964, 10.1029/JD093iD12p15955, 1988.

Diner, D., Abdou, W., Ackerman, T., Crean, K., Gordon, H., Kahn, R., Martonchik, J., McMuldroch, S., Paradise, S., Pinty, B., Verstraete, M., Wang, M., and West, R.: MISR level 2 aerosol retrieval algorithm theoretical basis, JPL D-11400, Rev. G, Jet Propul. Lab., Calif. Inst. of Technol., Pasadena, CA, USA, online available at: eospso.gsfc.nasa.gov/eos_homepage/for scientists/atbd, 2008.

Diner, D., Beckert, J., Reilly, T., Bruegge, C., Conel, J., Kahn, R., Martonchik, R., Ackerman, T., Davies, R., Gerstl, S., Gordon, H., Muller, J., Myneni, R., Sellers, P., Pinty, B., and Verstraete, M.: Multi-angle Imaging Spectro Radiometer (MISR) instrument description and experiment overview, IEEE Transactions on Geoscience and Remote Sensing, 36, 1072-1087, 1998.

Marey, H., Gille, J., El-Askary, H., Shalaby, E., and El-Raey, M.: Aerosol climatology over Nile Delta based on MODIS, MISR and OMI satellite data, Atmospheric Chemistry and Physics, 11, 10637-10648, 10.5194/acp-11-10637-2011, 2011.

Torres, O., Tanskanen, A., Veihelmann, B., Ahn, C., Braak, R., Bhartia, P., Veefkind, P., and Levelt, P.: Aerosols and surface UV products from Ozone Monitoring Instrument observations: An overview, Journal of Geophysical Research-Atmospheres, 112, D24S47, 10.1029/2007JD008809,

2007.

8. *L97, discussion of dust transport to the TP is originally discussed by Huang et al., 2007 and further supported by Xia et al. (2008), with regarding long-range transport from south Asia, some references should be added including Xia et al. (2011); Lu et al. (2012); Zhao et al. (2015). A detailed discussion on this issue can be found in a overview paper (Xia et al., 2021, AR).*

We agree with the reviewer and more references including that suggested here have been added, such as:

Cong, Z., Kang, S., Kawamura, K., Liu, B., Wan, X., Wang, Z., Gao, S., and Fu, P.: Carbonaceous aerosols on the south edge of the Tibetan Plateau: concentrations, seasonality and sources, Atmospheric Chemistry and Physics, 15, 1573-1584, 10.5194/acp-15-1573-2015, 2015.

Huang, J., Minnis, P., Yi, Y., Tang, Q., Wang, X., Hu, Y., Liu, Z., Ayers, K., Trepte, C., and Winker, D.: Summer dust aerosols detected from CALIPSO over the Tibetan Plateau, Geophysical Research Letters, 34, 10.1029/2007gl029938, 2007.

Lu, Z., Streets, D., Zhang, Q., and Wang, S.: A novel back-trajectory analysis of the origin of black carbon transported to the Himalayas and Tibetan Plateau during 1996-2010, Geophysical Research Letters, 39, 10.1029/2011gl049903, 2012.

Lüthi, Z., Skerlak, B., Kim, S., Lauer, A., Mues, A., Rupakheti, M., and Kang, S.: Atmospheric brown clouds reach the Tibetan Plateau by crossing the Himalayas, Atmospheric Chemistry and Physics, 15, 6007-6021, 10.5194/acp-15-6007-2015, 2015.

Xia, X., Zong, X., Cong, Z., Chen, H., Kang, S., and Wang, P.: Baseline continental aerosol over the central Tibetan plateau and a case study of aerosol transport from South Asia, Atmospheric Environment, 45, 7370-7378, 10.1016/j.atmosenv.2011.07.067, 2011.

Zhao, Z., Cao, J., Shen, Z., Xu, B., Zhu, C., Chen, L. W. A., Su, X., Liu, S., Han, Y., Wang, G., and Ho, K.: Aerosol particles at a high-altitude site on the Southeast Tibetan Plateau, China: Implications for pollution transport from South Asia, Journal of Geophysical Research-Atmospheres, 118, 11360-311375, 10.1002/jgrd.50599, 2013.

Zhu, J., Xia, X., Che, H., Wang, J., Cong, Z., Zhao, T., Kang, S., Zhang, X., Yu, X., and Zhang, Y.: Spatiotemporal variation of aerosol and potential long-range transport impact over the Tibetan Plateau, China, Atmospheric Chemistry and Physics, 19, 14637-14656, 10.5194/acp-19-14637-2019, 2019.

9. *L182, pls add standard deviation to the mean value.*

Thanks for helping figure this out. We have calculated the standard deviation of the annual mean AOD value, and related description has also been added at Lines 194-196: **"the annual average AODs over the Arctic, Antarctic, and the inner region of the TP are 0.046, 0.024, and 0.098 with the standard deviations of 0.003, 0.002, and 0.009, respectively."**

10. *L190-195, it seems transport of dust to the TP mainly occurs in summer, on the other hand, transport of fine particles from South Asia mainly occurs in dry season.*

We agree with the reviewer and the corresponding description has been modified at Lines 205-213: **"The aerosol loading over the TP is easily affected by the surrounding regions where there are many anthropogenic and natural aerosol sources. Specifically,**

**the dust aerosols in the Tarim Basin and Qaidam Basin have a greater contribution to the TP in spring and summer, especially in the northern part of the TP in summer (Huang et al., 2007; Xia et al., 2008; Xu et al., 2020). Meanwhile, a large number of fine aerosol particles exist in South Asia and the northern Indian Peninsula due to forest fires and anthropogenic burning during the dry season. The aerosols are lifted and transported to the Himalayas under the influence of large-scale atmospheric systems such as the South Asian monsoon and the Siberian high, which affects the southern part of the TP (Cong et al., 2015; Engling et al., 2011; Han et al., 2020; Xu et al., 2014; 2015)."**

References:

Huang, J., Minnis, P., Yi, Y., Tang, Q., Wang, X., Hu, Y., Liu, Z., Ayers, K., Trepte, C., and Winker, D.: Summer dust aerosols detected from CALIPSO over the Tibetan Plateau, Geophysical Research Letters, 34, 10.1029/2007gl029938, 2007.

Xia, X., Wang, P., Wang, Y., Li, Z., Xin, J., Liu, J., and Chen, H.: Aerosol optical depth over the Tibetan Plateau and its relation to aerosols over the Taklimakan Desert, Geophysical Research Letters, 35, 10.1029/2008gl034981, 2008.

Xu, C., Ma, Y., You, C., and Zhu, Z.: The regional distribution characteristics of aerosol optical depth over the Tibetan Plateau, Atmospheric Chemistry and Physics, 15, 12065-12078, 10.5194/acp-15-12065-2015, 2015.

Xu, X., Wu, H., Yang, X., and Xie, L.: Distribution and transport characteristics of dust aerosol over Tibetan Plateau and Taklimakan Desert in China using MERRA-2 and CALIPSO data, Atmospheric Environment, 237, 10.1016/j.atmosenv.2020.117670, 2020.

Cong, Z., Kang, S., Kawamura, K., Liu, B., Wan, X., Wang, Z., Gao, S., and Fu, P.: Carbonaceous aerosols on the south edge of the Tibetan Plateau: concentrations, seasonality and sources, Atmospheric Chemistry and Physics, 15, 1573-1584, 10.5194/acp-15-1573-2015, 2015.

Engling, G., Zhang, Y., Chan, C., Sang, X., Lin, M., Ho, K., Li, Y., Lin, C., and Lee, J.: Characterization and sources of aerosol particles over the southeastern Tibetan Plateau during the Southeast Asia biomass-burning season, Tellus Series B-Chemical and Physical Meteorology, 63, 117-128, 10.1111/j.1600-0889.2010.00512.x, 2011.

Han, H., Wu, Y., Liu, J., Zhao, T., Zhuang, B., Wang, H., Li, Y., Chen, H., Zhu, Y., Liu, H., Wang, Q. g., Li, S., Wang, T., Xie, M., and Li, M.: Impacts of atmospheric transport and biomass burning on the inter-annual variation in black carbon aerosols over the Tibetan Plateau, Atmospheric Chemistry and Physics, 20, 13591-13610, 10.5194/acp-20-13591-2020, 2020.

Xu, C., Ma, Y., Panday, A., Cong, Z., Yang, K., Zhu, Z., Wang, J., Amatya, P., and Zhao, L.: Similarities and differences of aerosol optical properties between southern and northern sides of the Himalayas, Atmospheric Chemistry and Physics, 14, 3133-3149, 10.5194/acp-14-3133-2014, 2014.

---

## Author Comment (AC2) · 21 Feb 2021

We thank the reviewers for their thoughtful, valuable, and detailed comments and suggestions that have helped us improve the paper quality. Our detailed responses (Blue) to the reviewers' questions and comments (*Italic*) are listed below.

Reviewer 2:

General comments: The manuscript entitled "Aerosol Characteristics in the Three Poles of the Earth Observed by CALIPSO" focuses on aerosol characteristics of the Arctic, Antarctic, and Tibetan Plateau by using CALIPSO L3 data and HYSPLIT model.

The results show that the AODs over three regions have obvious spatial and temporal feature. Different type of aerosol has remarkable spatial and seasonal. Overall, this manuscript is clear and well written. Some concerns are needed to address.

We highly appreciate the positive evaluation about our study along with the valuable comments that have helped us improve the paper quality a lot.

1. Page 1, Line 18-19, the sentence should be "The annual average AODs over the Arctic, Antarctic, and TP are 0.046.....", or "The annual mean values of AOD over the Arctic, Antarctic, and TP are 0.046.....".

The sentence has been corrected as suggested.

2. What is the basis of selecting the simulation points of backward trajectory? I recommend authors to add a simulation point over the eastern TP?

We are sorry for this confusion; the selection of back trajectory simulation sites is kind of subjective. The main selection basis is the relatively uniform distribution of sites so that the simulation results can retrieve the source of air masses in the study area.

Based on this suggestion, a simulation site over the eastern TP have been added, and the corresponding results are analyzed and added at Lines 444-449: "Similar to the site on the northern slope of the TP, the back trajectories of air masses at the eastern slope site are greatly affected by the Tarim Basin and Qaidam Basin in spring, but are mainly affected by the Iranian Plateau and the western part of the TP in autumn and winter. Differently, in summer, the back trajectories of air masses are not only from the Tarim Basin and Qaidam Basin, but also from the southern part of the TP with about 27.54% of air masses."

Figure 8: The seasonal climatological characteristics of the back trajectories (January 2007 to

December 2019) at each of the Arctic, Antarctic, and TP sites, separated by spring, summer, autumn, and winter.

3. Page 7, Line 178, the authors described "decrease with the increase of latitude in any season", however, Figure 2e shows some areas with high AODs at high latitudes of Antarctic in spring.

We appreciate this careful comment, and have corrected this sentence at Lines 188-191: "In general, aerosol loadings are found larger in the southern part of the Atlantic Ocean in the Antarctic and decrease with the increase of latitude, while high AODs could exist in some regions at high latitudes of the Antarctic such as the Antarctic Peninsula, especially in spring and winter."

4. Page 7, Line 190, "high aerosol concentrations in the Arctic and Antarctic mainly occur in winter and spring", The aerosol concentration of the Arctic should be higher in late fall to early spring, not just winter and spring (Figure 3).

Thanks for helping figure this out. We have changed the corresponding description and added at Lines 203-205: **"The high aerosol concentration mainly occurs from late autumn to early spring in the Arctic, while in winter and spring in the Antarctic."**

5. Page 7, Line 174, "the AOD averaged between Jun 2006 and Dec 2019"; Page 7, Line 187, "the monthly variations of multi-year (June 2016 - December 2019) average AODs"; Page 8, Lines 207-208, "the monthly AODs along with their standard deviations from June 2007 to December 2019". Why not use the AOD data for same period?

We highly appreciate these careful comments and feel sorry for our writing mistakes. We actually used the same period of June 2006 - December 2019. We have corrected them in the revised version.

6. Page 8, Lines 219-221, "First, there are anthropogenic emission sources over the TP region. Second, the TP is located in Central Asia surrounded by highly polluted areas, which is easily affected by external aerosol transport". The corresponding evidence or reference is needed. Following this suggestion, we added scientific references to support our descriptions in this section including

- Hu, Z., Huang, J., Zhao, C., Jin, Q., Ma, Y., and Yang, B.: Modeling dust sources, transport, and radiative effects at different altitudes over the Tibetan Plateau, Atmospheric Chemistry and Physics, 20, 1507-1529, 10.5194/acp-20-1507-2020, 2020.
- Li, C., Bosch, C., Kang, S., Andersson, A., Chen, P., Zhang, Q., Cong, Z., Chen, B., Qin, D., and Gustafsson,
  Ö.: Sources of black carbon to the Himalayan–Tibetan Plateau glaciers, Nature Communications, 7, 12574, 10.1038/ncomms12574, 2016.
- Liu, Y., Sato, Y., Jia, R., Xie, Y., Huang, J., and Nakajima, T.: Modeling study on the transport of summer dust and anthropogenic aerosols over the Tibetan Plateau, Atmos. Chem. Phys., 15, 12581-12594, 10.5194/acp-15-12581-2015, 2015.
- Zhao, C., Yang, Y., Fan, H., Huang, J., Fu, Y., Zhang, X., Kang, S., Cong, Z., Letu, H., and Menenti, M.: Aerosol characteristics and impacts on weather and climate over the Tibetan Plateau, National Science Review, 7, 492-495, 10.1093/nsr/nwz184, 2020.
- Zhu, J., Xia, X., Che, H., Wang, J., Cong, Z., Zhao, T., Kang, S., Zhang, X., Yu, X., and Zhang, Y.: Spatiotemporal variation of aerosol and potential long-range transport impact over the Tibetan Plateau, China, Atmospheric Chemistry and Physics, 19, 14637–14656, 10.5194/acp-19-14637-2019, 2019.

7. Page 8, Lines 219-221, "the wind speed in winter half-year in the Arctic region is higher than that in summer half-year". However, in Lines 199-200, "On the other hand, stable atmospheric status with less precipitation occurs in the Arctic winter". What is the real situation? It needs a verified evidence from satellite or reanalysis data.

We are sorry for the confusion. The wind speed in Lines 219-221 represents the wind speed near the surface, while the atmospheric stability in Lines 199-200 represents the vertical thermal stability. In order to avoid confusion, we changed the wind speed to the near-surface wind speed. In addition, we used the ERA-5 reanalysis data to investigate the atmospheric conditions over the Arctic region to support our descriptions at Lines 218-220 "As shown in Figure S1, the Arctic region has a smaller monthly average convective available potential energy (CAPE) in winter half-year, while the monthly average wind speed at 10 m above the surface is higher."